# Validation and Update of OMI Total Column Water Vapor Product

Huiqun Wang[1], Gonzalo Gonzalez Abad[1], Xiong Liu[1], Kelly Chance[1]

[1]Smithsonian Astrophysical Observatory, Cambridge, Massachusetts, 02421, USA

*Correspondence to*: Huiqun Wang (hwang@cfa.harvard.edu)

## 1 Introduction

Water vapor is an important factor for the weather and climate. It is the most abundant greenhouse gas and can amplify the effect of other greenhouse gases through positive feedback. It can condense to form clouds that greatly influence the heating rate and circulation of the atmosphere. In addition, it can influence atmospheric composition through photochemical reactions. Water vapor is highly variable in space and time. Better knowledge of its distribution is highly beneficial for predicting the weather, monitoring the climate and understanding the physics and chemistry of the atmosphere.

Water vapor has been observed using a variety of in-situ and remote-sensing techniques. Satellite remote sensing of water vapor has led to products retrieved from the visible (e.g., GOME (Wagner et al., 2003, Lang et al., 2007), SCIAMACHY (Noël et al., 2005), GOME-2 (Grossi et al., 2015), OMI (Wang et al., 2014)), near infrared (e.g., SCIAMACHY (Schrijver et al., 2009), MODIS (Diedrich et al., 2015), MERIS (Lindstrot et al., 2012)), infrared (e.g., MODIS (Seemann et al., 2003), AIRS (Bedka et al., 2010), IASI (Pougatchev et al., 2009)), microwave (e.g., SSM/I (Schlüssel et al., 1990; Wentz, 1997)), and GPS radio signals (e.g., Wang et al., 2007, Kishore et al., 2011). These datasets offer the unique opportunity to study water vapor distribution on a global scale. Among them, microwave and GPS measurements can be made under all-sky conditions. Other types of measurements are strongly affected by clouds. Infrared measurements can provide vertical profiles, but have low sensitivity to the planetary boundary layer where most water vapor resides. Microwave measurements are only available over non-precipitating ice-free ocean. Near infrared measurements have better quality over land. Visible measurements are available for both land and ocean but are limited to nearly cloud-free daytime conditions.

Wang et al. (2014) derived Total Column Water Vapor (TCWV, also known as the Total Precipitable Water, TPW) using the spectra measured by the Ozone Monitoring Instrument (OMI). The Level 2 data for 2005 – 2009 generated using the Wang et al. (2014) algorithm (Version 1.0) have been archived at the Aura Validation Data Center (AVDC). A detailed

assessment of data quality is important for data usage in various weather and climate studies. In
this paper, we perform a comprehensive validation of this product using the ground-based GPS
data from National Center for Atmospheric Research (NCAR), the near infrared sun-photometer
data from Aerosol Robotic Network (AERONET), and the microwave radiometer data from
Remote Sensing System (RSS). An updated OMI retrieval algorithm is also presented. The new
results are compared against RSS's microwave radiometer data and GlobVapour's
SSMI+MERIS data. The data sets used in this study are introduced in Section 2. The validation
of the Version 1.0 OMI data is performed in Section 3. The algorithm update is presented in
Section 4. A summary is provided in Section 5.

## 42 2 Total Column Water Vapor (TCWV) Data

### 43 2.1 OMI Data

OMI is an ultraviolet / visible (UV/VIS) imaging spectrometer on board the NASA EOS-
Aura satellite. It has three spectral channels spanning the 264 nm – 504 nm spectral region at 0.4
– 0.6 nm spectral resolution (Levelt et al., 2006). OMI has been providing daily global
observations at 13:45 LT with a nominal spatial resolution of 13 km $\times$ 24 km at nadir since July
48 2004.

Water vapor exhibits several distinct spectral bands in the OMI visible channel (349 nm –
504 nm). These features are several orders of magnitude weaker than those at longer
wavelengths. However, they can still be exploited to retrieve TCWV (Wagner et al., 2013; Wang
et al., 2014). Since water vapor is a weak absorber in the blue spectral range, the retrieval is free
from the complication of non-linearity due to saturation. Since the surface albedo is more
uniform over the globe in this spectral region, the signals do not change abruptly between land
and ocean. Water vapor derived from the blue spectral range can greatly enhance the scientific
return of satellites, especially for those with instruments that lack spectral coverage at longer
wavelengths (e.g., OMI).
Wang et al. (2014) retrieved TCWV from OMI spectra using the 430 nm – 480 nm retrieval
window. The retrieval method consists of two steps. First, the Slant Column Density (SCD) is
derived from a spectral fitting algorithm that considers water vapor, $O_3$, $NO_2$, $O_2$-$O_2$, liquid
water, $C_2H_2O_2$, the Ring effect, the water Ring effect, 3rd order closure polynomials, wavelength
shift, under-sampling and common mode. The median SCD fitting uncertainty is about 11%
(Wang et al., 2014). Then, the Vertical Column Density (VCD) is obtained by dividing the SCD
with an Air Mass Factor (AMF) that is based on a radiative transfer calculation. Wang et al.
(2014) found that the AMF was insensitive to wavelength, but sensitive to surface albedo and
highly sensitive to clouds. The albedo used in the AMF calculation is from an updated version of
the OMLER climatology at $0.5° \times 0.5°$ spatial resolution (Kleipool et al., 2008). The cloud
fraction and cloud top pressure used in the AMF calculation are from the second release of
Version 003 Level 2 OMCLDO2 product which is derived from the $O_2$-$O_2$ absorption band near
477 nm (Acarreta et al., 2004; Stammes et al., 2008). The VCD in molecules / $cm^2$ can be
converted to TCWV in mm using a multiplicative factor of $2.989 \times 10^{-22}$. The Collection 3
Version 1.0 Level 2 OMI water vapor data from 2005 to 2009 have been released at the AVDC
website (avdc.gsfc.nasa.gov). These data are validated in Section 3.
It should be noted that there are artificial stripes in the Level 2 OMI water vapor data. These
stripes are due to systematic errors related to instrument calibration. They can be smoothed by
post-processing Level 2 data. One smoothing method is to divide each line of SCD with a one-
dimensional (1D) smoothing array (Wang et al., 2014). As an example, the smoothing array as a
function of cross-track pixel number is shown for July 2005 (black) and July 2009 (gray) in
Figure 1. It is calculated from the monthly average of Level 2 SCDs and normalized using a $3^{rd}$
order polynomial fit (as a function of cross-track pixel number). The SCDs used are filtered to
pass the main data quality check (MDQFL = 0), have Root Mean Squared (RMS) fitting error <
$5 \times 10^{-3}$ and cloud fraction < 0.05. The MDQFL criterion checks that the fitting has converged,
the retrieved SCD is $< 4 \times 10^{23}$ molecules/$cm^2$ and the SCD is positive within $2\sigma$ fitting
uncertainty. Figure 1 shows a large pixel-to-pixel variation of up to 17%. Consequently, the
stripes in OMI Level 2 data can significantly influence comparisons with other datasets on a
daily timescale. OMI began to experience row anomalies since June 2007
(projects.knmi.nl/omi/research/product/rowanomaly-background.php). The affected rows in July
2009 are highlighted by dots in Figure 1. They appear to be more oscillatory than those in July
2005. However, the standard deviation of the smoothing array only increases from 6% for July
2005 to 7% for July 2009. While the data affected by OMI row anomaly should be used with
caution, their variation does not seem to be much larger than before, at least until July 2009.
Another smoothing method is to subtract a 1D offset array (as a function of pixel number)
from the SCD before its conversion to VCD. The offset array can be derived from a reference
region, such as the Sahara. The mean SCD of each cross-track pixel in the reference region is
calculated using the swaths obtained within a week, a low-order (e.g., third order) polynomial is
subsequently removed, and the resulting 1D array is used as the offset array. Since the smoothing
procedure is non-unique and can potentially introduce an additional bias, we use the un-
smoothed Level 2 OMI data (with stripes) in this paper.
**2.2 NCAR's Ground-based GPS Data**
NCAR hosts a 2-hourly TCWV dataset derived from the ground-based GPS measurements of
Zenith Path Delay (ZPD) at stations in the International GNSS Service (IGS), SuomiNet, and
GEONET networks (Wang et al., 2007). We have downloaded the data from
rda.ucar.edu/datasets/ds721.1/ (EOL/NCAR/UCAR, 2011, updated yearly). The IGS- SuomiNet
data include 1160 stations worldwide and are available from 1995 to 2012. The ground-based
GPS data have been extensively used to validate other TCWV measurements and data
assimilation products (Wang and Zhang, 2008, Sibelle et al., 2010, Mears et al., 2015). The GPS
TCWV retrieval error is estimated to be 1.5 mm (Wang et al., 2007). The mean difference
between the GPS and satellite microwave radiometer data over the ocean is < 1 mm and the
standard deviation is < 2 mm (Mears et al., 2015). In this paper, we use the subset of IGS-
SuomiNet data from 2005 to 2009 to compare with the OMI data.
**2.3 AERONET's Sun-photometer Data**
AERONET provides globally distributed observations of aerosol optical depth, TCWV, and
other variables using sun-photometers (Holben et al., 1998). The network has expanded from 16
sites in 1993 to 860 sites in 2014. TCWV is derived from the 940 nm filter that coincides with
the $2\nu_1+\nu_2$ water vapor absorption band. The Level 2.0 AERONET data are cloud screened and
quality assured (Smirnov et al., 2000). We have downloaded the publically available Version 2
Level 2.0 data from aeronet.gsfc.nasa.gov and used the subset from 2005 to 2009 to compare
with the OMI data. Using the subset of AERONET data observed at the sites operated by U.S.
Department of Energy Atmospheric Radiation Measurement (ARM) program, Pérez-Ramírez et
al. (2014) found that the AERONET TCWV had a general dry bias of $5 - 6\%$ and an estimated
uncertainty of $12 - 15\%$. The Version 3 AERONET data is currently under development and is
expected to be released in 2016.
**2.4 RSS's Microwave Data**
Remote Sensing Systems (RSS) generates TCWV data by processing the microwave data
from Special Sensor Microwave / Imager (SSM/I), Special Sensor Microwave Imager Sounder
(SSMIS), and other sensors. The retrieval uses a unified physically based algorithm which yields
a retrieval accuracy of 1.2 mm (Wentz, 1997). The TCWV data derived from these satellite
microwave radiometers are available under all-sky non-precipitating conditions over the ice-free
ocean. They have long been considered as among the most reliable and have been routinely
assimilated into numerical models. We have downloaded from www.remss.com the latest
Version 7 SSMIS data collected by the Defense Meteorological Satellite Program (DMSP)'s F16
satellite (Wentz et al., 2012). These data are obtained in both the morning (04:06 LT) and the
evening (16:06 LT), while OMI data are obtained in the early afternoon (13:45LT). The diurnal
cycle of TCWV varies with season and region and can sometimes exceeds 2 mm (Wang et al.,
2007). Abnormal conditions (heavy rain, sea ice, bad data, no observation, and land) are flagged
in the SSMIS data. In this paper, we make use of the daily gridded ($0.25° \times 0.25°$) SSMIS
product from 2005 to 2009.

**2.5 GlobVapour's SSMI+MERIS Data**

The GlobVapour project sponsored by the European Space Agency (ESA) Data User
Element (DUE) program generated a global Level 3 ($0.5° \times 0.5°$) TCWV product by combining
MERIS land and SSM/I ocean observations from 2003 to 2008 (www.globvapour.info). The
MERIS near IR data are collected around 10 AM and derived from the water vapor absorption
around 950 nm. The SSM/I microwave data are collected around 6 AM and derived using a 1D-
Var method for ice-free non-precipitating ocean. The GlobVapour Level 3 product combines
clear sky MERIS land data with all sky SSM/I ocean data. Over the land, GlobVapour is on
average about -1.3 mm lower than the GCOS Upper-Air Network (GUAN) radiosonde data and
+0.2 mm higher than the AIRS clear sky infrared data. Over the ocean, it is on average about
+1.3 mm higher than GUAN and +0.7 mm higher than AIRS. The standard deviation of the
difference ranges from 2 mm to 5 mm (Schröder and Bojkov, 2012). Wang et al. (2014)
compared the monthly mean GlobVapour data with the monthly mean Version 1.0 OMI data.
They found an overall agreement (within 1 mm) over land and an OMI low bias of -3 mm or
more over the ocean. In this paper, we sample the daily gridded GlobVapour data to compare
with the updated OMI data in Section 4.

# 3 V1.0 OMI TCWV Validation

**3.1 OMI and GPS**

The AVDC Collection 3 Level 2 OMI TCWV data processed with SAO's Version 1.0
algorithm are filtered and co-located with NCAR's ground-based GPS data. The filtering criteria
for OMI require that the general quality check is passed (MDQFL = 0), the cross-track quality
flag indicates that the retrieval is not affected by OMI's row anomaly, the SCD fitting RMS is <
$5 \times 10^{-3}$, the cloud fraction is < 10%, the cloud top pressure is > 500 hPa, and the AMF is > 0.75.
Since clouds can dramatically change the vertical sensitivities and lead to large errors in OMI
VCDs (Wang et al., 2014), the last three filtering criteria are intended to mitigate their influence.
These filtering criteria are also used in subsequent sections unless otherwise specified. Most of
the OMI data are filtered out due to cloud contamination. For July 1, 2005, there are about
1,255,000 data points satisfying the partial criteria of MDQFL = 0, no row anomaly and RMS <
$5 \times 10^{-3}$. Their average TCWV is 29.2 mm. Only about 30% of these data pass the full filtering
criteria, and their average is 21.7 mm. This suggests that clouds tend to increase the amount of
retrieved TCWV in OMI data.
For co-location at each GPS station, we select the GPS observations made between the local
noon and 14:00 LT each day. For each eligible GPS data point, we search the filtered OMI data
on the same day for the pixels that are within $0.25°$ latitude $\times 0.25°$ longitude of the GPS station.
For July 2005, co-located OMI data can be found for about half of the GPS observations. Among
them, there are typically around 4 (within a range of $1 - 16$) OMI data points for each GPS data
point. When multiple OMI data points are available for a single GPS data point, the average
weighted by the OMI SCD fitting error is calculated and used for comparison.
Figure 2 shows the TCWV time series comparison between the GPS and OMI data at
selected sites. These sites are scattered around the world (denoted by "X" in Figure 3) and
represent a variety of climate regimes. For both dry and wet conditions and for both small and
large seasonal cycles, the OMI data track the seasonal and inter-annual variations of the GPS
data well, even with the influence of stripes. This demonstrates the value of TCWV retrieved
from OMI.
The top panel of Figure 3 shows the (OMI – GPS) TCWV difference averaged within the
time period from January 1, 2005 to December 31, 2009 for the IGS-SuomiNet stations. For this
plot, we have excluded the stations with significant topography difference (i.e. those with
elevations that are different than the local gridded ($0.25°\times0.25°$) topography by 500 m or more).
We have also excluded the stations with < 100 data points. There are 250 stations in Figure 3.
Many are in North America and Europe, but very few are in Africa and on ocean islands.
Generally speaking, OMI data agree well with GPS data over land but are significantly lower
over the ocean. The histogram for the mean (OMI – GPS) TCWV difference (shown in the top
panel) is plotted in the bottom panel of Figure 3. It is binned by 0.5 mm and has a mode of -0.5
mm. OMI data agree with GPS data within 1.5 mm at 71% of the stations and within 3 mm at
89% of the stations. OMI data are higher than GPS data by 3 mm or more at 8 stations, where all
except for one station are located in coastal areas. OMI data are lower than GPS data by 3 mm or
more at 23 stations, where all except for 2 stations are located on ocean islands or in coastal
areas. OMI data are lower than GPS data by 5 mm or more at 10 stations, among which, 2
stations are located in coastal areas and the others are on ocean islands.
In the top row of Figure 4, we compare OMI with GPS TCWV using all available data pairs
at all land (left) and ocean (right) stations from 2005 to 2009. Since most GPS stations are over
land, the number of data points over land (317,118) far exceeds that over the ocean (2,621). The
data in the 2D histogram of OMI versus GPS are binned every 0.5 mm of TCWV. The largest
color-coded value in each panel is normalized to one. The GPS TCWV data over land are mostly
within the range of 4 mm (10% percentile) to 34 mm (90% percentile) and those over the ocean
are mostly within the range of 17 mm to 50 mm. The OMI data generally follow the GPS data
along the 1:1 line over land, but tend to be lower than the GPS data (i.e., below the 1:1 line) over
the ocean.
The middle row of Figure 4 shows the histograms for the (OMI – GPS) differences using the
data shown in the top row. The histogram for land stations has a peak at 0 mm. The distribution
is slightly asymmetric, with a Full-Width-at-Half-Maximum (FWHM) of 8.5 mm (from -5.0 mm
to 3.5 mm). The mean and median of the distribution are -0.3 mm and -0.4 mm, respectively.
The scatter is related to random errors in GPS data and random errors in OMI SCD, AMF, and
stripes. The histogram for the ocean stations is much less smooth due to the smaller sample size.
The distribution is apparently skewed towards more negative values and has a larger scatter. The
mode, mean, and median of (OMI – GPS) over the ocean are -1.5 mm, -3 mm, and -3.5 mm,
respectively.
The bottom row of Figure 4 shows the mean (cross, left axis), median (triangle, left axis) and
standard deviation (star, right axis) of (OMI – GPS) as functions of month for all the land (left)
and ocean (right) GPS stations. They are calculated using all the paired land (left) or ocean
(right) data for the corresponding month from 2005 to 2009. The number of data points used for
each month is about 20,000 – 30,000 for the land stations and only about 190 - 240 for the ocean
stations. For land stations, the median of (OMI – GPS) is close to 0 mm from December to May,
and becomes the most negative (around -1 mm) in July. The mean of (OMI – GPS) follows a
similar trend. The standard deviations vary between 4.8 mm and 7.1 mm, with a maximum in
August. For ocean stations, the sample size is much smaller. Nevertheless, results show larger
low biases for OMI. The means of (OMI – GPS) vary between -1 mm and -4 mm, and the
standard deviations vary between 8 mm and 11mm. The largest differences occur in June / July,
as do the standard deviations.

## 3.2 OMI and AERONET

We filter and co-locate OMI and AERONET TCWV data using the same procedure as that in
Section 3.1. Figure 5 shows time series comparisons at selected AERONET sites. These sites
represent a wide range of water vapor amounts and seasonal cycles around the world (denoted by
"X" in Figure 6). In general, OMI observations track the variations of AERONET data well
throughout 2005 - 2009. During the wet season, OMI data appear to be higher at several sites
(e.g., Skukuza, Mukdahan, GSFC, Hamburg, and Dakar).
In Figure 6, we examine the spatial distribution and histogram of the mean of (OMI –
AERONET) for the time period from 2005 to 2009. As in Figure 3, we have omitted the sites
with substantial topography difference and the sites with < 100 data points. Of the 160 stations
shown in Figure 6, there are only about 10 over the ocean. Figure 6 shows that OMI is generally

higher over land and lower over the ocean and in some coastal areas. The histogram shows a main peak at 0.5 mm and a secondary peak at -2.5 mm. The secondary peak is due to the ocean sites. 59% of the sites show a non-negative (OMI – AERONET) difference. Pérez-Ramírez et al. (2014) found a dry bias of AERONET TCWV at the US Southern Great Plains, Barrow (in Alaska) and Nauru islands (in the tropical western Pacific). Figure 6 suggests that OMI is slightly wetter than AERONET in the contiguous US and Alaska, but is even drier than AERONET at Nauru island.

In Figure 7, we compare OMI with AERONET TCWV using all data pairs from 2005 to 2009 at all land (left) and ocean (right) sites. The top row shows the 2D normalized histograms for OMI versus AERONET data and the middle row shows the histograms for (OMI – AERONET). Both are calculated using 0.5 mm bins. There are far more data points over land (91,350) than over the ocean (3,092). TCWV over the ocean is generally larger than that over land. The 10% and 90% percentiles of AERONET data for the ocean sites are 13 mm and 45 mm, while those for the land sites are 6 mm and 32 mm. Figure 7 shows that OMI generally agrees with AERONET well over land, but tends to be lower than AERONET over the ocean. The mean (median) of (OMI – AERONET) is 0 mm (-0.3 mm) for land and -2.0 mm (-2.6 mm) for the ocean. The (OMI-AERONET) histogram for land has a peak at -1 mm and an FWHM of 8.5 mm (from -5.0 mm to 3.5 mm), while that for the ocean has a peak at -3.5 mm and an FWHM of 12 mm (from -9.5 mm to 2.5 mm). The means, medians and standard deviations of (OMI – AERONET) as functions of month are shown in the bottom row for land (left) and ocean (right) sites. The mean of OMI agrees with that of AERONET within 0.3 mm over land, but is lower than AERONET by 0.6 mm to 2.4 mm over the ocean. These differences are a little smaller than those shown in Figure 4, which is consistent with a dry bias of AERONET TCWV reported by Pérez-Ramírez et al. (2014). The standard deviations of (OMI – AERONET) vary between 7 mm and 10 mm which are similar to those of (OMI – GPS).

**3.3 OMI and SSMIS**

The ground-based networks discussed before have poor coverage over the ocean, but the SSMIS TCWV data from RSS are specifically for the ocean and have long-term daily coverage. We will therefore use the SSMIS data as the reference for the ocean. In Figure 8, we compare the monthly mean OMI data (top row) with the monthly mean SSMIS data (middle row) for July 2005. The monthly gridded (0.25°×0.25°) OMI and SSMIS data are calculated from the monthly average of coincident daily gridded (0.25°×0.25°) Level 3 data.

The daily Level 3 SSMIS data are downloaded from RSS's website (www.remss.com). Both the morning and evening passes are used in the monthly average. Pixels with bad data and rain are filtered out. The resulting "all sky" data are associated with both clear sky and cloudy sky

conditions. In addition to water vapor column and rain rate, RSS's data also provide "cloud
liquid water path" for each pixel. In this paper, we use it to define a "clear" sky condition by
ignoring the pixels whose cloud liquid water path is > 0. Clouds in liquid phase are filtered out,
but ice clouds still remain. However, information for cloud ice is unavailable in the RSS data
used in this study. Therefore, the "clear" sky conditions referred to in this paper should be
considered as an approximation to cloud-free conditions.
The daily Level 3 OMI data are derived from the corresponding Level 2 data using the
average weighted by pixel area and slant column fitting uncertainty (Wang et al., 2014). The
selection criteria for gridding the OMI Level 2 data include MDQF = 0, no row anomaly, RMS <
$5\times10^{-3}$, AMF > 0.75, cloud top pressure > 500 mb, and cloud fraction < a cutoff value.
To compare with the "clear" sky monthly SSMIS data (second panel on the right of Figure
8), the OMI Level 2 data are gridded with a cloud fraction cutoff of 5% (first panel on the right).
Although a 0% cutoff is equivalent to the clear sky condition, we use a 5% cutoff here to retain
more data for gridding. The number of days when both OMI and SSMIS data are available at
each pixel is generally < 5 (third panel on the right). Nevertheless, it can be seen that OMI
captures the general spatial distribution of TCWV observed by SSMIS. However, OMI data tend
to be lower over the tropical oceans. The (OMI – SSMIS) difference has a global median of -4.7
mm and can be < -10 mm in the western Pacific and Atlantic. The difference between OMI and
"clear" sky SSMIS is smaller when a 10% cloud fraction cutoff is used (not shown), in which
case, the global median of (OMI – "clear" sky SSMIS) becomes -3.0 mm. However, the OMI
data quality is generally lower for cloudier scene as the AMF is highly sensitive to cloud (Wang
et al., 2014).
In the left column of Figure 8, we compare the monthly mean OMI and SSMIS data under all
sky conditions for July 2005. The monthly mean OMI data in the top left panel are calculated
from the daily gridded OMI data using a relaxed cloud fraction cutoff of 25%. This choice is
based on a balance between the cloudiness and the data quality for OMI. The monthly mean
SSMIS data in the second panel are calculated from the daily gridded all sky SSMIS data. Both
data sets are sampled and averaged in the same way. The number of data points used for monthly
averaging at each pixel (third panel) increases to >15 in most areas. Both the SSMIS (second
row) and the OMI (first row) data show increases in TCWV as cloud amount increases (from the
right to the left), but the increase is more pronounced in the OMI data. The (OMI – SSMIS)
difference (bottom row) is smaller for the all sky comparison than for the "clear" sky
comparison. Specifically, for the all sky case, the median difference becomes -1.7 mm, and the
difference becomes less negative in the western Pacific and Atlantic. There are some positive
values in the lower left panel. They are mostly located in areas of missing data in the lower right
panel, suggesting that the positive values are associated with significant cloud cover (5% – 25%).
This further indicates that the Version 1.0 OMI data tend to have a high bias under cloudy sky
conditions and a low bias under clear sky conditions. The cloudy sky high bias is mainly due to
the small AMF estimate, especially for clouds at high altitudes (not shown).
Figure 9 shows the same comparison as Figure 8, but for January 2005. Both OMI and
SSMIS data show the southward migration of the ITCZ from July to January and an increase of
TCWV with cloud fraction (from the right to the left in the top two rows). Again, the increase is
more pronounced for OMI than for SSMIS. For the "clear" sky comparison (right column), OMI
has a large low bias over the southern ocean, which can be -10 mm or more. The bias becomes
less negative and even positive for the all sky conditions, indicating that TCWV for the pixels
affected by clouds are higher for OMI than for SSMIS. The global median of (OMI – SSMIS) in
January 2005 is -6.5 mm for the "clear" sky comparison and -2.9 mm for the all sky comparison.
The top row of Figure 10 shows the 2D normalized histograms of Version 1.0 OMI versus
SSMIS for July 2005 (a, b) and January 2005 (c, d). The histograms are calculated using the
daily gridded (0.25°×0.25°) coincident data. The same OMI data filtering criteria as before are
applied except for a cloud fraction cutoff of 10%. This cutoff value is between the 5% and 25%
used in Figure 8 and Figure 9. We compare the OMI data with the "clear" sky SSMIS data in
Panel (a, c) and with the all sky SSMIS data in Panel (b, d). For each month, about 1 million
data points are used in the "clear" sky comparison and about 4 million in the all sky comparison.
Both the "clear" sky and the all sky results show that OMI is generally lower than SSMIS. The
(OMI – "clear" sky SSMIS) difference has a mean of -3.7 mm, a median of -3.7 mm, and a
standard deviation of 7.2 mm in July 2005. The difference is larger in January 2005, with a mean
of -4.9 mm, a median of -4.9 mm and a standard deviation of 7.1 mm. With the 10% cloud
fraction cutoff, the Version 1.0 OMI data are closer to the "clear" sky than to the all sky SSMIS
data, as the (OMI – all sky SSM/I) difference has a mean of -4.4 mm (-6.0 mm), a median of -4.3
mm (-6.0 mm), and a standard deviation of 7.7 mm (8.0 mm) in July (January) 2005.

## 4 Algorithm Update

### 4.1 SCD Fitting Update

The previous section shows that the AVDC Collection 3 Version 1.0 OMI data generally
agree well with the reference data over land but are lower over the ocean. This implies a bias in
the OMI SCD retrieval over the ocean. Wang et al. (2014) showed that liquid water is an
important molecule to consider in their retrieval algorithm. They found that the fitting residual is
generally larger over the ocean than over land. Moreover, the common mode derived over land
appears largely random, but that derived over the ocean has apparent spectral structures,
especially between 440 nm and 470 nm where the liquid water (Pope and Fry, 1997) and water
Ring reference spectrum exhibits distinct spectral features. Consequently, errors in liquid water
spectroscopy can lead to systematic errors in the water vapor retrieved over the ocean.
Furthermore, the 430 – 480 nm retrieval window used by Wang et al. (2014) contains both the $7\nu$
(435 – 450 nm) and the $6\nu+\delta$ (460 – 480 nm) spectral bands of water vapor. Lampel et al. (2015)
derived scaling factors for the water vapor absorption bands in the blue spectral range using the
$7\nu$ band as a reference. They suggested that the absorption strength of the $6\nu+\delta$ band should be
scaled by a factor of 1.02±0.07 in HITRAN 2008 (Rothman et al., 2009). This would also affect
the water vapor result derived from the 430 – 480 nm retrieval window.
To reduce the influence of errors in liquid water and water vapor cross sections, we have
experimented with narrower retrieval windows. With a narrower retrieval window, scaling of the
HITRAN water vapor spectrum can be avoided. Additionally, some broadband spectroscopy
error of liquid water can be accounted for by the $3^{rd}$ order closure polynomial. Using OMI orbit
5109, which cuts across the western Pacific on July 1, 2005, we varied the retrieval window
around the $7\nu$ water vapor band near 442 nm to maximize the retrieved median column amount
and minimize the median SCD fitting uncertainty. In addition, since water vapor over the ocean
is concentrated at the sea level, we have changed the water vapor reference spectra from one that
corresponds to 0.9 atm and 280K to one that corresponds to 1.0 atm and 288K. We recently
obtained the $O_2$-$O_2$ reference spectra measured by Thalman and Volkamer (2013). We therefore
updated it as well. All the other retrieval setups remain the same as those used in Version 1.0
(Wang et al., 2014).
The optimized new retrieval window is between 427.7 and 465.0 nm, using which, we obtain
a median VCD of $1.07\times10^{23}$ molecules/cm$^2$ and a median fitting uncertainty of $1.4\times10^{22}$
molecules/cm$^2$ for orbit 5109. We will refer to this retrieval algorithm as Version 2.0. For
comparison, the retrieval window of 430.0 – 460.0 nm leads to a median VCD of $1.01\times10^{23}$
molecules/cm$^2$ and a median uncertainty of $1.6\times10^{22}$ molecules/cm$^2$. For the same orbit, the
Version 1.0 algorithm leads to a median VCD of $8.6\times10^{22}$ molecules/cm$^2$ and a median
uncertainty of $1.1\times10^{22}$ molecules/cm$^2$. Although the absolute fitting uncertainty of the Version
2.0 algorithm is about 30% larger than that of Version 1.0, the median relative uncertainties of
both algorithms are about 12%.
The difference in TCWV between the Version 2.0 algorithm and the Version 1.0 algorithm
mainly comes from the change in retrieval window. With only the retrieval window change, the
median VCD of orbit 5109 increases from $8.6\times10^{22}$ molecules/cm$^2$ to $1.06\times10^{23}$ molecules/cm$^2$.
With a further change of the water vapor reference spectrum from 0.9 atm to 1.0 atm, the median
VCD increases to $1.07 \times 10^{23}$ molecules/cm$^2$. Updating the $O_2$-$O_2$ reference spectrum has a
negligible effect on the retrieval.
Using the Version 2.0 setup described above, we retrieved the Level 2 TCWV for July and
January 2005. Using the same method as that used in the top row of Figure 10, we generated
daily gridded Version 2.0 OMI data with a 10% cloud fraction cutoff and compared them with
the SSMIS daily gridded data in terms of the 2D histogram distributions in the bottom row of
Figure 10. The agreement between the Version 2.0 OMI and SSMIS data is much better than that
between the Version 1.0 OMI and SSMIS data. The low bias of the Version 1.0 OMI is
eliminated. For July 2005, the Version 2.0 OMI data follow the all sky SSMIS data along the 1:1
line well and are slightly higher than "clear" sky SSMIS data (by about 1 mm). For January, the
Version 2.0 OMI data follow the SSMIS data well when TCWV are below 20 mm, and are
slightly lower than the all sky SSMIS data for larger TCWV amount (by about 1 mm).
To investigate the spatial distribution of the changes between the Version 1.0 and Version 2.0
OMI data, we compare the monthly mean Level 3 gridded (0.25°×0.25°) data for July 2005. The
same filtering criteria as before have been applied. The top row of Figure 11 shows the (Version
2.0 OMI – Version 1.0 OMI) difference maps for a 5% (right) and a 25% (left) cloud fraction
cutoff. In both cases, the Version 2.0 OMI data increase slightly over land but substantially over
the ocean. Specifically, for the 5% cloud fraction cutoff, the Version 2.0 OMI data increase over
the Version 1.0 OMI data at AVDC by an average of 1.2 mm over land and 4.8 mm over the
ocean. For the 25% cutoff, the Version 2.0 OMI data increase by an average of 1.3 mm over land
and 3.7 mm over the ocean.
In the bottom row of Figure 11, we compare the Version 2.0 OMI data with the SSMIS data
for July 2005 using the same method as that for Figure 8. The bottom right panel shows the
result of (Version 2.0 OMI with a 5% cloud fraction cutoff – "clear" sky SSMIS). Comparing
with the bottom right panel of Figure 8, we find a better agreement here. Firstly, the previously
found large low bias (< -10 mm) of Version 1.0 OMI over the Pacific, Atlantic and Indian Ocean
is reduced by more than half. Secondly, the global mean difference decreases to 0.1 mm, which
is much smaller than before (-4.7 mm). Although the southern and northern mid / high latitudes
show some moderate positive values, these areas are affected by the small number of coincident
data points per pixel (3$^{rd}$ panel on the right of Figure 8). The bottom left panel of Figure 11
shows the difference of (Version 2.0 OMI with a 25% cutoff – all sky SSMIS). In comparison
with the bottom left panel of Figure 8, the Version 2.0 OMI data generally do not show any large
low bias. However, large high bias is seen in several places. As a result, the global mean over the
ocean change from -1.7 mm (Figure 8) to 2.9 mm (Figure 11). A comparison between the lower
left and lower right panel of Figure 11 reveals that these large positive values are consistently
located in the vicinity of the missing data of the lower right panel, which indicates that they are
affected by significant cloud cover. As discussed before, OMI cloudy data are expected to be less
reliable and tend to overestimate TCWV. This will partly compensate for any low bias if the
pixel is occasionally cloudy and show up as a high bias if the pixel is persistently cloudy.

## 4.2 AMF Update

AMFs are used to convert SCDs to VCDs. Consequently, errors in AMFs also affect OMI
TCWV. The AMFs in previous sections were derived by convolving the monthly mean water
vapor profiles used in the GEOS-Chem model ($2°\times2.5°$) with the scattering weights interpolated
from a look-up table (Wang et al., 2014). The look-up table was constructed using the radiative
transfer model VLIDORT (Spurr, 2006). The scattering weights in the look-up table depend on
surface pressure, surface albedo, Solar Zenith Angle (SZA), View Zenith Angle (VZA), Relative
Azimuth Angle (RAA), ozone column amount, cloud fraction, cloud pressure and wavelength.
The following updates have been made to the AMF calculation. (1) Using higher resolution
($0.5°\times0.5°$) a priori water vapor profiles generated by the MERRA-2 project of the Global
Modeling and Assimilation Office (GMAO). (2) Using the MERRA-2 surface pressure instead
of an estimate based on the surface topography and the 1976 US standard atmosphere. (3)
Reconstructing the look-up table with more reference points for surface albedo, cloud fraction
and cloud pressure, so that the interpolated values are more accurate. (4) Improving scattering
weight parameterization with respect to RAA. (5) Using simultaneously fitted ozone amounts in
scattering weight calculations. We will refer to the algorithm with both these AMF updates and
the SCD update described in Section 4.1 as Version 2.1.
We have retrieved TCWV using the Version 2.1 algorithm for July and January 2005. Figure
12 shows the result for July 2005. The OMI data used here correspond to a 5% cloud fraction
cutoff. The top left panel shows the monthly mean difference between Version 2.1 and Version
2.0 OMI data. The difference results from the AMF updates described above. Version 2.1 is
about 3 – 5 mm higher than Version 2.0 in the tropics, 3 – 5 mm lower over high topography,
and almost unchanged in other areas. The bottom left panel shows the monthly mean of (Version
2.1 OMI – "clear" sky SSMIS). It is calculated using the same method as that for the bottom
right panel of Figure 11. Comparing the two, we find a further reduction of the low bias over the
tropical oceans. In fact, the majority of the Version 2.1 OMI data between 0° and 30°N are now
within ±3 mm of the "clear" sky SSMIS data. The bottom right panel shows the histograms of
(OMI – "clear" sky SSMIS) for three versions of OMI retrievals. The mode of the distribution
shifts from -4.0 mm (Version 1.0) through 0 mm (Version 2.0) to 1.5 mm (Version 2.1). The top
right panel of Figure 12 shows the 2D normalized histogram of Version 2.1 OMI versus SSMIS
"clear" sky data. The slope is close to 1, but OMI is higher by about 1.5 mm, which is consistent
with the result shown in the bottom right panel.
In Figure 13 and Figure 14, we compare the Version 2.1 OMI data with the GlobVapour
MERIS+SSMI data for July and January 2005, respectively. The top left panel shows the
monthly mean of (OMI – GlobVapour). It is calculated as the average of coincident daily gridded
Level 3 data within the month. The OMI daily data are gridded with a 5% cloud fraction cutoff
to represent "clear" sky conditions. Note that GlobVapour's land data (MERIS) are for clear sky
conditions, but its ocean data (SSMI) are for all sky conditions. There are usually about 10 – 20
coincident data points / pixel in the low latitudes (upper right panel). The differences between
OMI and GlobVapour are generally within ±6 mm. Among them, large differences are typically
located in the areas where few data points exist, such as northern South America, central Africa,
eastern US, China and the Pacific rim in July. In areas with good statistics, the differences are
largely confined to within ±3 mm. The 2D normalized histograms of OMI versus GlobVapour
are shown in the middle row for land (left) and ocean (right). The two data sets follow each other
well. Over the ocean, OMI data are slightly higher than GlobVapour's SSMI data (by about 1
mm) in July and agree with GlobVapour's SSMI data in January. Over land, OMI data are
slightly higher than GlobVaour's MERIS data when TCWV is < 15 mm and slightly lower when
TCWV is > 15 mm. The normalized histograms of (OMI – GlobVapour) are shown in the
bottom row for land (left) and ocean (right). The distributions show that OMI agrees with
GlobVapour within ±1 mm for both land and ocean and for both July and January. The FWHM
of the histogram in July is 6 mm for both land and ocean, and that in January is 6 mm for ocean
and 1 mm for land.
**5 Summary**
The AVDC Collection 3 OMI TCWV data generated with the Version 1.0 algorithm are
compared with the NCAR's ground-based GPS network observations, AERONET's sun-
photometer observations and RSS's SSMIS microwave observations. Results show that the
AVDC OMI data track the seasonal and inter-annual variability of TCWV for a wide range of
climate regimes. The Version 1.0 OMI data agree well with other data sets over land, but show
significant low biases over the ocean. Over land, for all the available co-located data from
January 2005 to December 2009, (OMI – GPS) has a mean of -0.3 mm and a median of -0.4mm,
and (OMI – AERONET) has a mean of 0 mm and a median of -0.3 mm. Although (OMI - GPS
or AERONET) over land is larger in June – November than in December – April, even the
largest mean difference is between -1 mm and 0 mm. In comparison, over the ocean, the Version
1.0 OMI data (with cloud fraction < 5%) are on average lower than the "clear" sky SSMIS data
by about 4.7 mm in July 2005 and by about 6.5 mm in January 2005. The OMI low bias can be
greater than 10 mm over the western Pacific and Atlantic in July and over the southern ocean in
January. Clouds usually lead to large overestimates of OMI TCWV. As a result, the OMI data
with cloud fraction < 25% are significantly higher than the all sky SSMIS data in areas with
persistent cloud cover. We therefore do not recommend using OMI data that are affected by
clouds.
By reducing the retrieval window length from 430-480 nm to 427.7 – 465.0 nm and using the
water vapor reference spectra at the sea level, the Version 2.0 OMI algorithm can significantly
increase the retrieved TCWV over the ocean without affecting those over land much, leading to
better agreements with the reference datasets. For July 2005, the offset between the Version 2.0
OMI data (with cloud fraction < 5%) and the "clear" sky SSMIS data over the western Pacific
and Atlantic is reduced by more than half, and the global mean difference over the ocean
improves to 0.1 mm.
By updating the AMF calculations (Section 4.2) in addition to the SCD fitting, for July 2005
the Version 2.1 retrieval algorithm leads to a further reduction of the Version 2.0 OMI low bias
in the western Pacific and Atlantic and the mean of (Version 2.1 OMI – "clear" sky SSMIS)
becomes 1.5 mm. The Version 2.1 OMI data agree with GlobVapour's MERIS+SSMI data
within ±1 mm for both land and ocean and for both July and January 2005, although the
distribution's FWHM is 6 mm.
**Acknowledgement**
This paper is supported by NASA's Atmospheric Composition: Aura Science Team program
(Sponsor Contract Number NNX14AF56G). We thank NASA's Aura Validation Data Center
(avdc.gsfc.nasa.gov) for generating and archiving OMI water vapor product. The GPS
precipitable water data are downloaded from rda.ucar.edu/datasets/ds721.1. The AERONET
Version 2 total column water vapor data are downloaded from
aeronet.gsfc.nasa.gov/new_web/data.html. The SSMIS data are produced by Remote Sensing
Systems (RSS), sponsored by the NASA Earth Science MEaSUREs Program and are available at
www.remss.com. The GlobVapour MERIS+SSM/I data are downloaded from
www.globvapour.info.

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

 **Figure captions**

**Figure 1.** Smoothing array for stripes in Version 1.0 OMI TCWV as a function of cross-track
pixel number for July 2005 (black) and July 2009 (gray). The pixels affected by the row anomaly
are indicated by dots.

**Figure 2.** Time series comparison between the Version 1.0 OMI (red) and GPS (black) data at
selected GPS stations from January 1, 2005 to December 31, 2009.

**Figure 3.** (Top) Spatial distribution of the mean of (Version 1.0 OMI – GPS) from 2005 to 2009
at IGS-SuomiNet stations. (Bottom) Histogram (with 0.5 mm bins) for the values in the top
panel.

**Figure 4.** (Top) 2D normalized histograms for (left) land and (ocean) derived from all the paired
Version 1.0 OMI and GPS data at all suitable IGS-SuomiNet stations from January 1, 2005 to
December 31, 2009. Results are shown for 0.5 mm × 0.5 mm bins, with the largest binned value
normalized to 1. The black line in each panel corresponds to 1:1. (Middle) Histograms of (OMI –
GPS) derived from the same data as those used in the top panel. The counts correspond to 0.5
mm bins. (Bottom) Median (triangle, left axis), mean (cross, right axis), and standard deviation
(star, right axis) of (Version 1.0 OMI – GPS) as functions of month.

**Figure 5.** Time series comparison between Version 1.0 OMI (red) and AERONET (black) at
selected AERONET stations from January 1, 2005 to December 31, 2009.

**Figure 6.** (Top) Spatial distribution of the time mean of (Version 1.0 OMI – AERONET) from
2005 to 2009 at AERONET stations. (Bottom) Histogram (with 0.5 mm bins) for the values in
the top panel.

**Figure 7.** (Top) 2D normalized histograms for (left) land and (right) ocean derived from all the
paired Version 1.0 OMI and AERONET data at all suitable AERONET stations from January 1,
2005 to December 31, 2009. Results are shown for 0.5 mm × 0.5 mm bins, with the largest
binned value normalized to 1. The black line in each panel corresponds to 1:1. (Middle)
Histograms of (Version 1.0 OMI – AERONET) derived from the same data as those used in the
top row. The counts correspond to 0.5 mm bins. (Bottom) Mean (triangle, left axis), media
(cross, right axis), and standard deviation (star, right axis) of (Version 1.0 OMI – AERONET) as
functions of month for (left) land and (right) ocean sites.

**Figure 8.** (First row) Monthly mean Version 1.0 OMI TCWV (mm) for cloud fraction (left) <
25% and (right) <5% for July 2005. (Second row) Monthly mean SSMIS TCWV (mm) for July
2005 for (left) all sky and (right) "clear" sky conditions. (Third row) Number of coincident data
points per pixel within July 2005 for the corresponding column. (Fourth row) First row - second
row). White areas in the maps represent missing data.

**Figure 9.** The same as Figure 8, but for January 2005.

**Figure 10.** Two-dimensional normalized histograms derived from daily gridded (0.5° × 0.5°)
OMI (with cloud fraction < 10%) and SSMIS data using 0.5 mm × 0.5 mm bins. The black line
in each panel is the 1:1 line. (a) Version 1.0 OMI versus "clear" sky SSMIS for July 2005 (b)
Version 1.0 OMI versus all sky SSMIS for July 2005 (c) Version 1.0 OMI versus "clear" sky
SSMIS for January 2005 (d) Version 1.0 OMI versus all sky SSMIS for January 2005 (e)
Version 2.0 OMI versus "clear" sky SSMIS for July 2005 (f) Version 2.0 OMI versus all sky
SSMIS for July 2005 (g) Version 2.0 OMI versus "clear" sky SSMIS for January 2005 (h)
Version 2.0 OMI versus all sky SSMIS for January 2005.

**Figure 11.** (Top row) Monthly mean of (Version 2.0 OMI – Version 1.0 OMI) for cloud fraction
(left) < 25% and (right) < 5% for July 2005. (Bottom left) Monthly mean of (Version 2.0 OMI
with cloud fraction < 25% - all sky SSMIS) for July 2005. (Bottom right) Monthly mean of
(Version 2.0 OMI with cloud fraction < 5% - "clear" sky SSMIS) for July 2005.

**Figure 12.** (Top left) Version 2.1 – Version 2.0 monthly mean OMI with cloud fraction < 5% for
July 2005. The other three panels are composed using coincident daily gridded (0.5°×0.5°) OMI
(with cloud fraction < 5%) and "clear" sky SSMIS data for July 2005. (Bottom left) Monthly
mean of (Version 2.1 OMI– SSMIS). (Top right) 2D normalized histogram of Version 2.1 OMI
versus SSMIS composed using 0.5 mm × 0.5 mm TCWV bins. (Bottom right) Histogram of
(Version 1.0 OMI – SSMIS) in black, (Version 2.0 OMI – SSMIS) in blue and (Version 2.1 OMI
– SSMIS) in red.

**Figure 13.** Comparison between Version 2.1 OMI (with cloud fraction < 5%) and GlobVapour
data (1°×1°) for July 2005. All panels are composed using coincident daily gridded data. (Top
left) Monthly mean of (OMI – GlobVapour). White areas represent missing data. (Top right)
Number of coincident data points per pixel. (Middle row) 2D normalized histograms of Version
2.1 OMI versus GlobVapour for (left) land and (right) ocean. (Bottom row) Histograms of
(Version 2.1 OMI – GlobVapour) for (left) land and (right) ocean.

**Figure 14.** The same as Figure 13, but for January 2005.