# Peer review of "Validation and Update of OMI Total Column Water Vapor Product"

_Atmospheric Chemistry and Physics, 2016_

## Referee Comment (RC1) · Anonymous Referee #1 · 16 May 2016

**Validation of OMI Total Column Water Vapor Product**

by Huiqun Wang, Gonzalo Gonzalez Abad, Xiong Liu, and Kelly Chance

**General comments:**

In the manuscript "Validation of OMI Total Column Water Vapor Product" the authors compare the Collection 3 OMI Total Column Water Vapour product with the NCAR's ground based GPS data, the AERONET sun-photometer data and the Remote Sensing System's SSM/I data set. The knowledge of the global distribution of water vapour is fundamental for global atmospheric models aiming to predict weather and monitor climate and is well within the scope of the Atmospheric Chemistry and Physics journal. This is also an important contribution since it exploits the possibility of retrieving water vapour in the blue spectral range and the future Sentinel missions will all lack spectral coverage in the green to red part of the spectrum (above 500 nm)

Substantial conclusions are reached regarding the influence of liquid water on the large low bias observed over ocean and the necessity to reduce the retrieval window to improve the overall consistency of the data set. The 430 - 480 retrieval window used by Wang et al. contains two water vapour bands. By exploiting a smaller (427.7 – 465.0 nm) retrieval window optimized around the 7v band, the authors found a better agreement with the other data sets.

The results and explanations are sufficient to support the interpretation that the large fitting residuals observed over ocean depend on errors in liquid water spectroscopy and water vapour cross sections. Level 2 TCWV data obtained using a new setups are compared with the SSM/I data under different cloud conditions and clearly show an improved agreement. Moreover, the updates discussed in the paper will be considered in the next release of the Smithsonian Astrophysical Observatory OMI water vapour.

The paper is well organized and the validation results are presented in distinct and balanced sections for the different data sets: GPS, AERONET and SSM/I. After a description of the different products and filtering procedures used for the comparisons, the quality of the OMI data is investigated via time series comparisons, spatial distributions and histograms of the mean and median differences. However, the statistical significance of the results hasn't been deeply discussed. More detailed comments are addressed below.

**Detailed comments:**

Section 3: Comparison results: While discussing the filtering criteria for OMI, the authors empathize the importance of mitigating the cloud influence. It would be useful for the reader to add a more detailed description of the cloud treatment (derivation of cloud fraction and cloud pressure) and to discuss the effect of applying the filtering to the original product (TCWV statistics, global sampling along with other effects).

Section 3.3: OMI and SSM/I: The authors compare the monthly mean SSM/I and OMI data under 'all' and 'clear' sky conditions and obtain larger negative values in the latter case (for July 2005). However, as suggested by the results of Figure 9, the mean bias between daily gridded SSM/I and OMI data is slightly larger for the 'all' sky data (for July 2007). I would suggest to use only daily co-located measurements to perform the monthly comparisons (Figure 8). In fact, a large bias might arise from the different sampling in cloud-free and cloud-contaminated products when comparing Level 3 data sets.

Results concerning the comparison between the SSM/I and OMI old and new products are showed for July. Could you please extend your analysis also to other months or comment on the seasonality (if any) of the outcomes?

Section 4: Algorithm Update: The authors derived the 427.7 – 465.0 nm retrieval window by optimizing around the 7v water vapour band in the OMI spectra. Could you please further motivate this choice and discuss the sensitivity of water vapour to other retrieval window choices?

As shown in Wang et al. (2014), the fitting uncertainty for shorter retrieval window is larger, but the median SCDs decreases as the retrieval window length increases. How these results compare to the current analysis?

The authors state that the differences between the new algorithm and the Version 1.0.0 algorithm mainly come from the change in the retrieval window. Could you please quantify the effects of each modification to the original algorithm?

Although the proposed update to the retrieval algorithm goes in the direction of reducing the bias between the OMI product and the other data sets, residual uncertainties might arise from the AMF calculations, clouds treatment, aerosols, and so on... Could you please give a general comment on the performance of the algorithm (and in particular with respect to the AMF conversion)?

Summary: The new water vapour algorithm can significantly increase the retrieved TCWV over the ocean without affecting those over land much. It might be beneficial to compare the results obtained with the new algorithm with an independent global data set (for example ECMWF reanalysis data, GOME, SSM/I + MERIS...).

Figure 2, 5: The labels of the time series plot could be improved setting regular (yearly) time intervals.

Figure 3, 6 (top): I would suggest to use a discrete scale of coloring to improve readability.

---

## Referee Comment (RC2) · Anonymous Referee #2 · 10 Jun 2016

**Review Comments:**

**Validation of OMI Total Column Water Vapour Product**

**General comments:**

The manuscript "Validation of OMI Total Column Water Vapor Product" Describes the geophysical validation of Aura Validation Data Center (AVDC) Collection 3 OMI Total Column Water Vapour data set against ground-based GPS, AERONET sunphotometer and satellite-based SSM/I data over period from 2005 to 2009. Authors report good agreement against GPS and AERONET data over land and general underestimation against SSM/I over ocean. Manuscript then describes experimental setup for retrieval algorithm to improve retrieval over ocean and presents comparisons between new algorithm and AVDC TCWV data.

Subject matter is well suited to ACP and methods used are sound and explained clearly. Figures shown are meaningful, although their clarity and quality should be improved. Interpretation of the results shown in section 3 is somewhat lacking and results should be contrasted with other published work. More detailed look on effects of the measurement parameters (viewing geometry, surface albedo, seasonal variation etc.) would also improve the article. However, the manuscript is sufficiently sound to be published in ACP after revision.

**Detailed Comments:**

Section 3.: Does the general quality screening (MDQF=0) include screening for OMI row anomaly?

Section 3.1: While it is good idea to conduct these comparisons for cloud free cases, it would be good to show the effect of the cloud cover on reliability of the observations. Are cloudy observations still useful?

Section 3.3: Results here should be contrasted to other published work. How does the OMI TCWV compare against other TCWV products (as mentioned in introduction), when they are compared to SSM/I?

Figures 2. and 5.: Is there a change in long-term levels due to the changes in viewing geometry? Due to asymmetrical nature of the row anomaly, any dependencies on viewing geometry (SZA, VZA, local time?) in the product might affect the long term time series even after screening.

Figures 4. and 7.: What is the reason for overestimation over ocean for large total water vapour columns, especially in contrast to general underestimation over ocean? Are there specific situations where OMI TCWV is not reliable?

All figures: Please clarify the figures, especially colour scales used.

---

## Author Comment (AC1)

**Response to Comment on**
**"Validation of OMI Total Column Water Vapor Product"**

by Huiqun Wang, Gonzalo Gonzalez Abad, Xiong Liu, and Kelly Chance

Thank you very much for your review. We have revised our manuscript accordingly. Please find our response to each comment below. To facilitate reading, we have highlighted your review in blue with Arial font, our response in black with Arial font, and our revised text in black with Times New Roman font.

**General comments:**

In the manuscript "Validation of OMI Total Column Water Vapor Product" the authors compare the Collection 3 OMI Total Column Water Vapour product with the NCAR's ground based GPS data, the AERONET sun-photometer data and the Remote Sensing System's SSM/I data set. The knowledge of the global distribution of water vapour is fundamental for global atmospheric models aiming to predict weather and monitor climate and is well within the scope of the Atmospheric Chemistry and Physics journal. This is also an important contribution since it exploits the possibility of retrieving water vapour in the blue spectral range and the future Sentinel missions will all lack spectral coverage in the green to red part of the spectrum (above 500 nm)

Substantial conclusions are reached regarding the influence of liquid water on the large low bias observed over ocean and the necessity to reduce the retrieval window to improve the overall consistency of the data set. The 430 - 480 retrieval window used by Wang et al. contains two water vapour bands. By exploiting a smaller (427.7 – 465.0 nm) retrieval window optimized around the 7v band, the authors found a better agreement with the other data sets. The results and explanations are sufficient to support the interpretation that the large fitting residuals observed over ocean depend on errors in liquid water spectroscopy and water vapour cross sections. Level 2 TCWV data obtained using a new setups are compared with the SSM/I data under different cloud conditions and clearly show an improved agreement. Moreover, the updates discussed in the paper will be considered in the next release of the Smithsonian Astrophysical Observatory OMI water vapour.

The paper is well organized and the validation results are presented in distinct and balanced sections for the different data sets: GPS, AERONET and SSM/I. After a description of the different products and filtering procedures used for the comparisons, the quality of the OMI data is investigated via time series comparisons, spatial distributions and histograms of the mean and median differences. However, the statistical significance of the results hasn't been deeply discussed. More detailed comments are addressed below.

**Detailed comments:**

Section 3: Comparison results: While discussing the filtering criteria for OMI, the authors emphathize the importance of mitigating the cloud influence. It would be useful for the reader to add a more detailed description of the cloud treatment (derivation of cloud fraction and cloud pressure) and to discuss the effect of applying the filtering to the original product (TCWV statistics, global sampling along with other effects).

The cloud fraction and cloud top pressure we used to calculate the AMF are from the OMCLDO2 product which is downloaded from mirador.gsfc.nasa.gov. Detailed information for the cloud retrieval algorithm can be found in Acarreta et al. (2004) and Stammes et al. (2008).

In the third paragraph of Section 2.1, we have added "The cloud fraction and cloud top pressure used in the AMF calculation are from the second release of Version 003 Level 2 OMCLDO2 product which is derived from the $O_2$-$O_2$ absorption band near 477 nm (Acarreta et al., 2004; Stammes et al., 2008)."

For the effect of filtering, in the first paragraph of Section 3.1, we have added "Most of the OMI data are filtered out due to cloud contamination. For July 1, 2005, there are about 1,255,000 data points satisfying the partial criteria of MDQFL = 0, no row anomaly and RMS $< 5 \times 10^{-3}$. Their average TCWV is 29.2 mm. Only about 30% of these data pass the full filtering criteria, and their average is 21.7 mm. This suggests that clouds tend to increase the amount of retrieved TCWV in OMI data."

In the second paragraph of Section 3.1, we have added "For July 2005, co-located OMI data can be found for about half of the GPS observations. Among them, there are typically around 4 (within a range of 1 – 16) OMI data points for each GPS data point."

Section 3.3: OMI and SSM/I: The authors compare the monthly mean SSM/I and OMI data under 'all' and 'clear' sky conditions and obtain larger negative values in the latter case (for July 2005). However, as suggested by the results of Figure 9, the mean bias between daily gridded SSM/I and OMI data is slightly larger for the 'all' sky data (for July 2007). I would suggest to use only daily co-located measurements to perform the monthly comparisons (Figure 8). In fact, a large bias might arise from the different sampling in cloud-free and cloud-contaminated products when comparing Level 3 data sets.

Following your suggestion, we have remade Figure 8 so that the daily co-located measurements are used to calculate the monthly average. We have added two panels to the figure showing the number of coincident data points per pixel for the corresponding cases. The revised Figure 8 is attached to the interative comment file. The reference data used are actually RSS's SSMIS data collected by F16 instead of SSM/I. We have made this correction in the paper. Compared with the "clear" sky SSMIS data, the Version 1.0 OMI data (with cloud fraction < 5%) show large low biases over the western Pacific and Atlantic. The difference between the Version 1.0 OMI data (with cloud fraction < 25%) and the all sky SSMIS data is smaller. However, OMI shows a large high bias for pixels affected by significant cloud cover. We have revised the corresponding text in Section 3.3 to the following.

" The ground-based networks discussed before have poor coverage over the ocean, but the SSMIS TCWV data from RSS are specifically for the ocean and have long-term daily coverage. We will therefore use the SSMIS data as the reference for the ocean. In Figure 8, we compare the monthly mean OMI data (top row) with the monthly mean SSMIS data (middle row) for July 2005. The monthly gridded ($0.25° \times 0.25°$) OMI and SSMIS data are calculated from the monthly average of coincident daily gridded ($0.25° \times 0.25°$) Level 3 data.

The daily Level 3 SSMIS data are downloaded from RSS's website (www.remss.com). Both the morning and evening passes are used in the monthly average. Pixels with bad data and rain are filtered out. The resulting "all sky" data are associated with both clear sky and cloudy sky conditions. In addition to water vapor column and rain rate, RSS's data also provide "cloud liquid water path" for each pixel. In this paper, we use it to define a "clear" sky condition by ignoring the pixels whose cloud liquid water path is > 0. Clouds in liquid phase are filtered out, but ice clouds still remain. However, information for cloud ice is unavailable in the RSS data used in this study. Therefore, the "clear" sky conditions referred to in this paper should be considered as an approximation to cloud-free conditions.

The daily Level 3 OMI data are derived from the corresponding Level 2 data using the average weighted by pixel area and slant column fitting uncertainty (Wang et al., 2014). The selection criteria for gridding the OMI Level 2 data include MDQF = 0, no row anomaly, RMS < $5 \times 10^{-3}$, AMF > 0.75, cloud top pressure > 500 mb, and cloud fraction < a cutoff value.

To compare with the "clear" sky monthly SSMIS data (second panel on the right of Figure 8), the OMI Level 2 data are gridded with a cloud fraction cutoff of 5% (first panel on the right). Although a 0% cutoff is equivalent to the clear sky condition, we use a 5% cutoff here to retain more data for gridding. The number of days when both OMI and SSMIS data are available at each pixel is generally < 5 (third panel on the right). Nevertheless, it can be seen that OMI captures the general spatial distribution of TCWV observed by SSMIS. However, OMI data tend to be lower over the tropical oceans. The (OMI – SSMIS) difference has a global median of -4.7 mm and can be < -10 mm in the western Pacific and Atlantic. The difference between OMI and "clear" sky SSMIS is smaller when a 10% cloud fraction cutoff is used (not shown), in which case, the global median of (OMI – "clear" sky SSMIS) becomes -3.0 mm. However, the OMI data quality is generally lower for cloudier scene as the AMF is highly sensitive to cloud (Wang et al., 2014).

In the left column of Figure 8, we compare the monthly mean OMI and SSMIS data under all sky conditions for July 2005. The monthly mean OMI data in the top left panel are calculated from the daily gridded OMI data using a relaxed cloud fraction cutoff of 25%. This choice is based on a balance between the cloudiness and the data quality for OMI. The monthly mean SSMIS data in the second panel are calculated from the daily gridded all sky SSMIS data. Both data sets are sampled and averaged in the same way. The number of data points used for monthly averaging at each pixel (third panel) increases to >15 in most area. Both the SSMIS (second row) and the OMI (first row) data show an increase in TCWV as cloud amount increases (from the right to the left), but the increase is more pronounced in the OMI data. The (OMI – SSMIS) difference (bottom row) is smaller for the all sky comparison than for the "clear" sky comparison. Specifically, for the all sky case, the

median difference becomes -1.7 mm, and the difference becomes less negative in the western Pacific and Atlantic. There are some positive values in the lower left panel. They are mostly located in areas of missing data in the lower right panel, suggesting that the positive values are associated with significant cloud cover (5% – 25%). This further indicates that the Version 1.0 OMI data tend to have a high bias under cloudy sky condition and a low bias under clear sky condition."

Results concerning the comparison between the SSM/I and OMI old and new products are showed for July. Could you please extend your analysis also to other months or comment on the seasonality (if any) of the outcomes?

We have expanded the comparison by adding Figure 9 which is the same as Figure 8 but for January 2005. We have updated the names of subsequent figures accordingly. We have also revised the original Figure 9 (now Figure 10) so that it includes results for both July and January. These figures are attached to the interative comments filefile. The following paragraph is added to the end of Section 3.3.

"     Figure 9 shows the same comparison as Figure 8, but for January 2005. Both OMI and SSMIS data show the southward migration of the ITCZ from July to January and an increase of TCWV with cloud fraction (from the right to the left in the top two rows). Again, the increase is more pronounced for OMI than for SSMIS. For "clear" sky comparison (right column), OMI has a large low bias over the southern ocean which can be -10 mm or more. The bias becomes less negative and even positive for all sky conditions, indicating that TCWV for the pixels affected by clouds are higher for OMI than for SSMIS. The global median of (OMI – SSMIS) in January 2005 is -6.5 mm for the "clear" sky comparison and -2.9 mm for the all sky comparison.

    The top row of Figure 10 shows the 2D normalized histograms of Version 1.0 OMI versus SSMIS for July 2005 (a, b) and January 2005 (c, d). The histograms are calculated using the daily gridded (0.25°×0.25°) coincident data. The same OMI data filtering criteria as before are applied except for a cloud fraction cutoff of 10%. This cutoff value is between the 5% and 25% used in Figure 8 and Figure 9. We compare the OMI data with the "clear" sky SSMIS data in Panel (a, c) and with the all sky SSMIS data in Panel (b, d).  For each month, about 1 million data points are used in the "clear" sky comparison and about 4 million in the all sky comparison. Both the "clear" sky and the all sky results show that OMI is generally lower than SSMIS. The (OMI – "clear" sky SSMIS) difference has a mean of -3.7 mm, a median of -3.7 mm, and a standard deviation of 7.2 mm in July 2005. The difference is larger in January 2005, with a mean of -4.9 mm, a median of -4.9 mm and a standard deviation of 7.1 mm. With the 10% cloud fraction cutoff, the Version 1.0 OMI data are closer to the "clear" sky than to the all sky SSMIS data, as the (OMI – all sky SSM/I) difference has a mean of -4.4 mm (-6.0 mm), a median of -4.3 mm (-6.0 mm), and a standard deviation of 7.7 mm (8.0 mm) in July (January) 2005."

    We have also added panels showing the monthly mean, median and standard deviation of (Version 1.0 OMI – GPS) in Figure 4 and (Version 1.0 OMI – AERONET) in Figure 7. The following discussion is added to the corresponding sections.

For Figure 4, "     The bottom row of Figure 4 shows the mean (cross, left axis), median (triangle, left axis) and standard deviation (star, right axis) of (OMI – GPS) as functions of month for all the land (left) and ocean (right) GPS stations. They are calculated using all the paired land (left) or ocean (right) data for the corresponding month from 2005 to 2009. The number of data points used for each month is about 20,000 – 30,000 for the land stations and only about 190 - 240 for the ocean stations. For land stations, the median of (OMI – GPS) is close to 0 mm from December to May, and becomes the most negative (around -1 mm) in July. The mean of (OMI – GPS) follows a similar trend. The standard deviations vary between 4.8 mm and 7.1 mm, with a maximum in August. For ocean stations, the sample size is much smaller. Nevertheless, results show larger low biases for OMI. The mean of (OMI – GPS) vary between -1 mm and -4 mm, and the standard deviations vary between 8 mm and 11mm. The largest differences occur in June / July, so do the standard deviations."

For Figure 7, "The mean, median and standard deviation of (OMI – AERONET) as functions of month are shown in the bottom row for land (left) and ocean (right) sites. The mean of OMI agrees with that of AERONET within 0.3 mm over land, but is lower than AERONET by 0.6 mm to 2.4 mm over the ocean. These differences are a little smaller than those shown in Figure 4, which is consistent with a dry bias of AERONET TCWV reported by Pérez-Ramírez et al. (2014). The standard deviations of (OMI – AERONET) vary between 7 mm and 10 mm which are similar to those of (OMI – GPS)."

Section 4: Algorithm Update: The authors derived the 427.7 – 465.0 nm retrieval window by optimizing around the 7v water vapour band in the OMI spectra. Could you please further motivate this choice and discuss the sensitivity of water vapour to other retrieval window choices? As shown in Wang et al. (2014), the fitting uncertainty for shorter retrieval window is larger, but the median SCDs decreases as the retrieval window length increases. How these results compare to the current analysis?

In the 2$^{nd}$ paragraph of Section 4.1, we have added "With a narrower retrieval window, scaling of the HITRAN water vapor spectrum can be avoided. Additionally, some broadband spectroscopy error of liquid water can be accounted for by the 3$^{rd}$ order closure polynomial. Using OMI orbit 5109 which cuts across the western Pacific on July 1, 2005, we varied the retrieval window around the 7v water vapor band near 442 nm to maximize the retrieved medium column amount and minimize the medium SCD fitting uncertainty."

We have added the following paragraph after the 2$^{nd}$ paragraph of Section 4.1.

"The optimized new retrieval window is between 427.7 and 465.0 nm, using which, we obtain a medium VCD of $1.07{\times}10^{23}$ molecules/cm$^2$ and a medium fitting uncertainty of $1.4{\times}10^{22}$ molecules/cm$^2$ for Orbit 5109. We will refer to this retrieval algorithm as Version 2.0. For comparison, the retrieval window of 430.0 – 460.0 nm leads to a medium VCD of $1.01{\times}10^{23}$ molecules/cm$^2$ and a medium uncertainty of $1.6{\times}10^{22}$ molecules/cm$^2$. For the same orbit, the Version 1.0 algorithm leads to a medium VCD of $8.6{\times}10^{22}$ molecules/cm$^2$ and a medium uncertainty of $1.1{\times}10^{22}$ molecules/cm$^2$. Although the absolute fitting uncertainty of the Version 2.0 algorithm is about 30% larger than that of Version 1.0, the medium relative uncertainties of both algorithms are about 12%."

The authors state that the differences between the new algorithm and the Version 1.0.0 algorithm mainly come from the change in the retrieval window. Could you please quantify the effects of each modification to the original algorithm?

We have added the following paragraph in Section 4 to quantify the effects of the modification.

" The difference in TCWV between the Version 2.0 algorithm and the Version 1.0 algorithm mainly comes from the change in retrieval window. With only the retrieval window change, the medium VCD of orbit 5109 increases from $8.6{\times}10^{22}$ molecules/cm$^2$ to $1.06{\times}10^{23}$ molecules/cm$^2$. With a further change of the water vapor reference spectrum from 0.9 atm to 1.0 atm, the medium VCD increases to $1.07{\times}10^{23}$ molecules/cm$^2$. Updating the $O_2$-$O_2$ reference spectrum has a negligible effect on the retrieval."

Although the proposed update to the retrieval algorithm goes in the direction of reducing the bias between the OMI product and the other data sets, residual uncertainties might arise from the AMF calculations, clouds treatment, aerosols, and so on... Could you please give a general comment on the performance of the algorithm (and in particular with respect to the AMF conversion)?

We have added Section 4.2 (below) to describe the AMF updates (in addition to SCD retrieval updates) and their effect. Figure 12, 13 and 14 are new figures included in this section. They are attached to the interative comments file. Wang et al. (2014) examined the sensitivity of AMF to surface albedo, cloud information, viewing geometry and wavelength. We are in the process of investigating the AMF error due to various dependent variables for each retrieval and plan to include AMF error estimate in our future data release. For consistency with the OMCLDO2 product and for computational efficiency, our current AMF calculation does not consider aerosols. We hope to get a better understanding of the effect of aerosol on OMI water vapor retrieval through a dedicated future study.

"4.2 AMF Update

AMFs are used to convert SCDs to VCDs. Consequently, errors in AMFs also affect OMI TCWV. The AMFs in previous sections were derived by convolving the monthly mean water vapor profiles used in the GEOS-Chem model (2°×2.5°) with the scattering weights interpolated from a look-up table (Wang et al., 2014). The look-up table was constructed using the radiative transfer model VLIDORT (Spurr, 2006). The scattering weights in the look-up table depend on surface pressure, surface albedo, Solar Zenith Angle (SZA), View Zenith Angle (VZA), Relative Azimuth Angle (RAA), ozone column amount, cloud fraction, cloud pressure and wavelength.

The following updates have been made to the AMF calculation. (1) Using higher resolution (0.5°×0.5°) a priori water vapor profiles generated by the MERRA-2 project of the Global Modeling and Assimilation Office (GMAO). (2) Using the MERRA-2 surface pressure instead of an estimate based on the surface topography and the 1976 US standard air. (3) Reconstructing the look-up table with more reference points for surface albedo, cloud fraction and cloud pressure, so that the interpolated values are more accurate. (4) Improving scattering weight parameterization with respect to RAA. (5) Using simultaneously fitted ozone amounts to calculate scattering weights. We will refer to the algorithm with both these AMF updates and the SCD update described in Section 4.1 as Version 2.1.

We have retrieved TCWV using the Version 2.1 algorithm for July and January 2005. Figure 12 shows the result for July 2005. The OMI data used here correspond to a 5% cloud fraction cutoff. The top left panel shows the monthly mean difference between Version 2.1 and Version 2.0 OMI data. The difference results from the AMF updates described above. Version 2.1 is about 3 – 5 mm higher than Version 2.0 in the tropics, 3 – 5 mm lower over high topography, and almost unchanged in other areas. The bottom left panel shows the monthly mean of (Version 2.1 OMI – "clear" sky SSMIS). It is calculated using the same method as that for the bottom right panel of Figure 11. Comparing the two, we find a further reduction of the low bias over the tropical oceans. In fact, the majority of the Version 2.1 OMI data between 0° and 30°N are now within ±3 mm of the "clear" sky SSMIS data. The bottom right panel shows the histograms of (OMI – "clear" sky SSMIS) for three versions of OMI retrievals. The mode of the distribution shifts from -4.0 mm (Version 1.0) through 0 mm (Version 2.0) to 1.5 mm (Version 2.1). The top right panel of Figure 12 shows the 2D normalized histogram of Version 2.1 OMI versus SSMIS "clear" sky data. The slope is close to 1, but OMI is higher by about 1.5 mm which is consistent with the result shown in the bottom right panel.

In Figure 13 and Figure 14, we compare the Version 2.1 OMI data with the GlobVapour MERIS+SSMI data for July and January 2005, respectively. The top left panel shows the monthly mean of (OMI – GlobVapour). It is calculated as the average of coincident daily gridded Level 3 data within the month. The OMI daily data are gridded with a 5% cloud fraction cutoff to represent "clear" sky conditions. Note that GlobVapour's land data (MERIS) are for clear sky conditions, but its ocean data (SSMI) are for all sky conditions. There are usually about 10 – 20 coincident data points / pixel in the low latitudes (upper right panel). The differences between OMI and GlobVapour are generally within ±6 mm. Among them, large differences are typically located in the areas where few data points exist, such as northern South America, central Africa, eastern US, China and the Pacific rim in July. In areas with good statistics, the differences are largely confined to within ±3 mm. The 2D normalized histograms of OMI versus GlobVapour are shown in the middle row for land (left) and ocean (right). The two data sets follow each other well. Over the ocean, OMI data are slightly higher than GlobVapour's SSMI databy about 1 mm in July and agrees with GlobVapour's SSMI data in January. Over land, OMI data are slightly higher than GlobVapour's MERIS data when TCWV is < 15 mm and slightly lower when TCWV is > 15 mm. The normalized histograms of (OMI – GlobVapour) are shown in the bottom row for land (left) and ocean (right). The distributions show that OMI agrees with GlobVapour within ±1 mm for both land and ocean and for both July and January. The FWHM in July is 6 mm for both land and ocean, and that in January is 6 mm for ocean and 1 mm for land."

Summary: The new water vapour algorithm can significantly increase the retrieved TCWV over the ocean without affecting those over land much. It might be beneficial to compare the results obtained with the new algorithm with an independent global data set (for example ECMWF reanalysis data, GOME, SSM/I + MERIS...).

We have added comparisons between the updated (Version 2.1) OMI and GlobVapour's MERIS+SSM/I data in Figure 13 and Figure 14 in Section 4.

Figure 2, 5: The labels of the time series plot could be improved setting regular (yearly) time intervals.

We have improved the labels of Figure 2 and Figure 5. The revised figures are attached to the interactive comments file.

Figure 3, 6 (top): I would suggest to use a discrete scale of coloring to improve readability.

We have changed the color scheme of Figure 3 and Figure 6. The revised figures are attached to the interactive comments file. Please find all the figure captions below.

**Figure Captions**

[revised manuscript text omitted]

---

## Author Comment (AC2)

**Response to Review Comments on**

**"Validation of OMI Total Column Water Vapour Product"**

Thank you very much for your review. We have revised our manuscript accordingly. Please find our response to each comment below. To facilitate reading, we have highlighted your review in blue with Arial font, our response in black with Arial font, and our revised text in black with Times New Roman font.

**General comments:**

The manuscript "Validation of OMI Total Column Water Vapor Product" Describes the geophysical validation of Aura Validation Data Center (AVDC) Collection 3 OMI Total Column Water Vapour data set against ground-based GPS, AERONET sunphotometer and satellite-based SSM/I data over period from 2005 to 2009. Authors report good agreement against GPS and AERONET data over land and general underestimation against SSM/I over ocean. Manuscript then describes experimental setup for retrieval algorithm to improve retrieval over ocean and presents comparisons between new algorithm and AVDC TCWV data.

Subject matter is well suited to ACP and methods used are sound and explained clearly. Figures shown are meaningful, although their clarity and quality should be improved. Interpretation of the results shown in section 3 is somewhat lacking and results should be contrasted with other published work. More detailed look on effects of the measurement parameters (viewing geometry, surface albedo, seasonal variation etc.) would also improve the article. However, the manuscript is sufficiently sound to be published in ACP after revision.

**Detailed Comments:**

Section 3.: Does the general quality screening (MDQF=0) include screening for OMI row anomaly?

MDQF = 0 does not check for OMI row anomaly. However, unless specified otherwise, we filter out the pixels affected by row anomaly using the cross-track data quality flag. This has now been clarified.

Towards the end of Section 2.1, we added "The MDQFL criterion checks that the fitting has converged, the retrieved SCD is $< 4 \times 10^{23}$ molecules/cm$^2$ and the SCD is positive within 2σ fitting uncertainty."

We modified the first paragraph of Section 3.1 to "…The filtering criteria for OMI require that the general quality check is passed (MDQFL = 0), the cross-track quality flag indicates that the retrieval is not affected by OMI's row anomaly, the SCD fitting RMS is $< 5 \times 10^{-3}$, the cloud fraction is $< 10\%$, the cloud top pressure is $> 500$ hPa, and the AMF is $> 0.75$".

Section 3.1: While it is good idea to conduct these comparisons for cloud free cases, it would be good to show the effect of the cloud cover on reliability of the observations. Are cloudy observations still useful?

When we compare Version 1.0 OMI data with RSS's SSMIS data in Section 3.3, we found that cloudy OMI data have a large high bias associated with small AMF estimate. We do not recommend using cloudy OMI data. The following changes are made.

In Section 3.3, when we discuss Figure 8, "There are some positive values in the lower left panel. They are mostly located in areas of missing data in the lower right panel, suggesting that the positive values are associated with significant cloud cover (5% – 25%). This further indicates that the Version 1.0 OMI data tend to have a high bias under cloudy sky condition and a low bias under clear sky condition. The cloudy sky high bias is mainly due to the small AMF estimate, especially for high clouds (not shown)."

In Section 5, "OMI data (with cloud fraction $< 25\%$) are significantly higher than all sky SSMIS data in areas with persistent cloud cover. We therefore do not recommend using OMI data that are affected by clouds".

Section 3.3: Results here should be contrasted to other published work. How does the OMI TCWV compare against other TCWV products (as mentioned in introduction), when they are compared to SSM/I?

Wang et al. (2014) published comparisons between Version 1.0 OMI and GlobVapour's MERIS+SSMI data. In this revision, we have added Section 4.2 (below) that includes comparisons between Version 2.1 OMI and GlobVapour's MERIS+SSMI data (Figure 13 and Figure 14, attached to the interactive comments file). We use GlobVapour since its validation document contains extensive comparisons with other data sets. We have added Section 2.5 to introduce GlobVapour's MERIS+SSMI data.

**"2.5 GlobVapour's SSMI+MERIS Data**

The GlobVapour project sponsored by the European Space Agency (ESA) Data User Element (DUE) program generated a global Level 3 (0.5°×0.5°) TCWV product by combining MERIS land and SSM/I ocean observations from 2003 to 2008 (www.globvapour.info). The MERIS near IR data are collected around 10 AM and derived from the water vapor absorption around 950 nm. The SSM/I microwave data are collected around 6 AM and derived using a 1D-Var method for ice-free non-precipitating ocean. The GlobVapour Level 3 product combines clear sky MERIS land data with all sky SSM/I ocean data. Over the land, GlobVapour is on average about -1.3 mm lower than the GCOS Upper-Air Network (GUAN) radiosonde data and +0.2 mm higher than the AIRS clear sky infrared data. Over the ocean, it is on average about +1.3 mm higher than GUAN and +0.7 mm higher than AIRS. The standard deviation of the difference ranges from 2 mm to 5 mm (Schröder and Bojkov, 2012). Wang et al. (2014) compared the monthly mean GlobVapour data with the monthly mean Version 1.0 OMI data. They found an overall agreement (within 1 mm) over land and an OMI low bias of -3 mm or more over the ocean. In this paper, we sample the daily gridded GlobVapour data to compare with the updated OMI data in Section 4."

"4.2 AMF Update

AMF is used to convert SCD to VCD. Consequently, errors in AMF also affect OMI TCWV. The AMFs in previous sections were derived by convolving the monthly mean water vapor profiles used in the GEOS-Chem model (2°×2.5°) with the scattering weights interpolated from a look-up table (Wang et al., 2014). The look-up table was constructed using the radiative transfer model VLIDORT (Spurr, 2006). The scattering weights in the lookup table depend on surface pressure, surface albedo, Solar Zenith Angle (SZA), View Zenith Angle (VZA), Relative Azimuth Angle (RAA), ozone column amount, cloud fraction, cloud pressure and wavelength.

The following updates have been made to the AMF calculation. (1) Using higher resolution (0.5°×0.5°) a priori water vapor profiles generated by the MERRA-2 project of the Global Modeling and Assimilation Office (GMAO). (2) Using the MERRA-2 surface pressure instead of an estimate based on the surface topography and the 1976 US standard air. (3) Reconstructing the look-up table with more reference points for surface albedo, cloud fraction and cloud pressure, so that the interpolated values are more accurate. (4) Improving scattering weight parameterization with respect to RAA. (5) Using simultaneously fitted ozone amount in scattering weight calculation. We will refer to the algorithm with both these AMF updates and the SCD update described in Section 4.1 as Version 2.1.

We have retrieved TCWV using the Version 2.1 algorithm for July and January 2005. Figure 12 shows the result for July 2005. The OMI data used here correspond to a 5% cloud fraction cutoff. The top left panel shows the monthly mean difference between Version 2.1 and Version 2.0 OMI data. The difference results from the AMF updates described above. Version 2.1 is about 3 – 5 mm higher than Version 2.0 in the tropics, 3 – 5 mm lower over high topography, and almost unchanged in other areas. The bottom left panel shows the monthly mean of (Version 2.1 OMI – "clear" sky SSMIS). It is calculated using the same method as that for the bottom right panel of Figure 11. Comparing the two, we find a further reduction of the low bias over the tropical oceans. In fact, the majority of the Version 2.1 OMI data between 0° and 30°N are now within ±3 mm of the "clear" sky SSMIS data. The bottom right panel shows the histograms of (OMI – "clear" sky SSMIS) for three versions of OMI retrievals. The mode of the distribution shifts from -4.0 mm (Version 1.0) through 0 mm (Version 2.0) to 1.5 mm (Version 2.1). The top right panel of Figure 12 shows the 2D normalized histogram of Version 2.1 OMI versus SSMIS "clear" sky data. The slope is close to 1, but OMI is higher by about 1.5 mm which is consistent with the result shown in the bottom right panel.

In Figure 13 and Figure 14, we compare the Version 2.1 OMI data with the GlobVapour MERIS+SSMI data for July and January 2005, respectively. The top left panel shows the monthly mean of (OMI – GlobVapour). It is calculated as the

average of coincident daily gridded Level 3 data within the month. The OMI daily data are gridded with a 5% cloud fraction cutoff to represent "clear" sky condition. Note that GlobVapour's land data (MERIS) are for clear sky condition, but its ocean data (SSMI) are for all sky condition. There are usually about 10 – 20 coincident data points / pixel in the low latitudes (upper right panel). The differences between OMI and GlobVapour are generally within ±6 mm. Among them, large differences are typically located in the areas where few data points exist, such as northern South America, central Africa, eastern US, China and the Pacific rim in July. In areas with good statistics, the differences are largely confined to within ±3 mm. The 2D normalized histograms of OMI versus GlobVapour are shown in the middle row for land (left) and ocean (right). The two data sets follow each other well. Over the ocean, OMI data are slightly higher than GlobVapour's SSMI data by about 1 mm in July and agrees with GlobVapour's SSMI data in January. Over land, OMI data are slightly higher than GlobVapour's MERIS data when TCWV is < 15 mm and slightly lower when TCWV is > 15 mm. The normalized histograms of (OMI – GlobVapour) are shown in the bottom row for land (left) and ocean (right). The distributions show that OMI agrees with GlobVapour within ±1 mm for both land and ocean and for both July and January. The FWHM in July is 6 mm for both land and ocean, and that in January is 6 mm for ocean and 1 mm for land."

Figures 2. and 5.: Is there a change in long-term levels due to the changes in viewing geometry? Due to asymmetrical nature of the row anomaly, any dependencies on viewing geometry (SZA, VZA, local time?) in the product might affect the long term time series even after screening.

We have not noticed any apparent long-term change associated with viewing geometry from 2005 to 2009 in Figures 2 and 5.

Figures 4. and 7.: What is the reason for overestimation over ocean for large total water vapour columns, especially in contrast to general underestimation over ocean? Are there specific situations where OMI TCWV is not reliable?

The large overestimation of OMI TCWV over the ocean is due to cloud contamination, as it is much more apparent with the 25% cloud fraction cutoff than with the 5% cloud fraction cutoff (Figure 8). The SCD fitting uncertainty of the cloudy pixels is not particularly larger than normal, however, clouds lead to an underestimation of AMF and therefore an overestimation of VCD.

In Section 3.3, when we discuss Figure 8, "There are some positive values in the lower left panel. They are mostly located in areas of missing data in the lower right panel, suggesting that the positive values are associated with significant cloud cover (5% – 25%). This further indicates that the Version 1.0 OMI data tend to have a high bias under cloudy sky condition and a low bias under clear sky condition. The cloudy sky high bias is mainly due to the small AMF estimate, especially for clouds at high altitudes (not shown)."

In Section 4.1, when we compare the updated Version 2.0 OMI data with RSS's SSMIS data, we have added "In comparison with the bottom left panel of Figure 8, the Version 2.0 OMI data generally do not show any large low bias. However, large high bias is seen in several places. As a result, the global mean over the ocean change from -1.7 mm (Figure 8) to 2.9 mm (Figure 11). A comparison between the lower left and lower right panel of Figure 11 reveals that these large positive values are consistently located in the vicinity of the missing data of the lower right panel, which indicates that they are affected by significant cloud cover. As discussed before, OMI cloudy data are expected to be less reliable and tend to overestimate TCWV. This will partly compensate for any low bias if the pixel is occasionally cloudy and show up as a high bias if the pixel is persistently cloudy."

In the summary, we have added "Clouds usually lead to large overestimates of OMI TCWV. As a result, the OMI data with cloud fraction < 25% are significantly higher than the all sky SSMIS data in areas with persistent cloud cover. We therefore do not recommend using OMI data that are affected by clouds."

All figures: Please clarify the figures, especially colour scales used.

We have revised our figures (attached to the interactive comments file). We notice that our .ps files have better quality than those shown in the WORD document which was subsequently converted to PDF file. We will upload the figures separately this time.